# OpenReview forum: "Open-Source Can Be Dangerous: On the Vulnerability of Value Alignment in Open-Source LLMs"
_ICLR.cc/2024/Conference — Submitted to ICLR 2024_

### Official Review · Reviewer_dUB6 · 2023-10-28

**Soundness:** 2 fair
**Presentation:** 2 fair
**Contribution:** 2 fair
**Rating:** 6
**Confidence:** 5

**Summary:**

This paper studies the risks implied by releasing model weights for capable and safety-tuned instruction-following language models.

In particular, the authors present two fine-tuning approaches which can revert the safety mechanisms built in state-of-the-art released open models. The first approach RSFT fine-tunes the model on harmful data. The second approach RVA applies direct preference optimization with the harmful response as the preferred entry in the pair.

**Strengths:**

There has been a lot of recent debate on the pros and cons of releasing model weights in the scientific community. The paper studies how safety-tuned models can be adapted to produce harmful content at a much higher frequency. This is a timely work which highlights the risks of releasing model weights and contributes a viewpoint to the open model debate.

**Weaknesses:**

There are several weaknesses. I list them below by category.

**Risks of model release**: Authors base their studies on safety-tuned models and present approaches that revert the safety mechanisms (e.g., RLHF). However, from a practical viewpoint, a raw pretrained model can likely be more easily adapted to produce harmful content than a safety-tuned model. Given that raw pretrained models are typically released along with safety-tuned variants (e.g., Llama2 / Llama2-Chat), it's worthwhile to measure how fine-tuning approaches work there. How much easier is it to adapt a raw pretrained model to produce harmful content compared to a safety-tuned one?

**Economic incentives**: The cost of performing these adversarial adaptations is likely low, considering the main method is LoRA fine-tuning on a small batch of examples. However, it's still useful to elucidate the cost of these attacks to make the analysis complete. How much does it cost ($) to adapt one model? How much time does it take? How many GPUs would you need?

**Clarity on harmful content**: Authors use the generic umbrella term "harmful" as the inverse term of "value-aligned". However, it is worthwhile to clarify the types of harmful content being studied. Is it primarily hate speech and toxicity? Or is it misinformation or something else?

I read the author comments and updated my score. The submission remains a borderline one given its limited novelty.

**Questions:**

- For GPT-4 evaluation, what quality control measures have the authors adopted? To what extent are the numbers trustworthy?
- MMLU is a useful proxy for measuring general model capability. But it is not a good proxy for assessing models' helpfulness in following instructions. Have the authors attempted other datasets for helpfulness evaluation? E.g., Anthropic's HH dataset or AlpacaEval?
- The title is misleading for two reasons:
  - "Open-source" is a very specific term which doesn't accurately describe the settings for which these attacks may be applied. For
  instance, the supervised fine-tuning attack can be applied to API models (OpenAI fine-tuning API). In addition, certain models have
  released weights but are not open-source -- the famous llama model is released under restricted licenses and thus doesn't count as
open-source. But the model itself can be called an open model.
  - Despite the drawbacks studied in the paper, open-source also has its benefits. For instance, this research is made possible with open models with weight releases. Thus, saying "open-source is dangerous" conveys a limited and inadequate view.

**Details Of Ethics Concerns:**

Authors should discuss more the long-term ethical implications of this work, extending the "implications for future work" section.

---

> ### Author Response · Authors · 2023-11-19
>
> **W1**: How much easier is it to adapt a raw pre-trained model to produce harmful content compared to a safety-tuned one?
>
> **A1**: Thanks for your comments. In Appendix A.2.4, we added experiments to reverse-align pre-trained model Llama2 (7B).
>
> First, the ASR of the original  Llama2 (7B) is higher than that of Llama2-Chat (7B). This is because Llama2 has not undergone alignment. In cases where Llama2 fails to answer harmful questions, it is often because the model has not undergone instruction tuning, resulting in incorrect execution of instructions, rather than a refusal to respond based on values.
>
> Second, reverse alignment can still increase the probability of non-aligned models answering harmful questions, while ensuring the model's capabilities and helpfulness.  The observations are quite natural and consistent with the statements in [1]. We would like to clarify that our work focuses more on value-aligned LLMs to investigate whether value alignment is a sufficient defense strategy for open-access LLMs and calls for more robust releasing strategies.
>
> [1] GPT-4 Technical Report.
>
> ---
>
> **W2**: Economic incentives: The cost of performing these adversarial adaptations is likely low, considering the main method is LoRA fine-tuning on a small batch of examples. However, it's still useful to elucidate the cost of these attacks to make the analysis complete. How much does it cost ($) to adapt one model? How much time does it take? How many GPUs would you need?
>
> **A2**: We provide computation time in Appendix A.4. Our solution is efficient since we use LoRA and a relatively short max length (i.e., 512). In our experiments, we use 8 V100s to reverse-align LLMs. For methods using prompt-prefix pairs, it only takes around 3 hours due to their short responses. We also notice that the experiments do not occupy all the GPU memories, indicating that we could potentially use fewer GPUs to achieve reverse alignment.
>
> ---
>
> **W3**: Clarity on harmful content: Authors use the generic umbrella term "harmful" as the inverse term of "value-aligned". However, it is worthwhile to clarify the types of harmful content being studied. Is it primarily hate speech and toxicity? Or is it misinformation or something else?
>
> **A3**: Thanks for your insightful comments. Our test dataset contains different types of harmful tasks, but we do not analyze the ASR of different types of attacks in our original paper. In Appendix A.2.1 of the revision, we analyze the ASRs of 13 types of attacks in ForbidQ. There are three observations from the analysis.
> 1. LLMs exhibit a certain level of bias in sensitivity to different attack types.
> 2. Different models have varying robustness.
> 3. The effectiveness of different methods varies.
>
> ---
>
> **Q1**: For GPT-4 evaluation, what quality control measures have the authors adopted? To what extent are the numbers trustworthy?
>
> **A1**: Thanks for your great comment. To verify the trustworthiness of GPT-4 evaluation, we randomly chose 200 samples to verify the consistency of the results between GPT-4 evaluation and human evaluation. 92.5% of GPT-4 judgments are consistent with human evaluation, which validates the effectiveness of GPT-4 evaluation.
>
> ---
>
> **Q2**: MMLU is a useful proxy for measuring general model capability. However, it is not a good proxy for assessing models' helpfulness in following instructions. Have the authors attempted other datasets for helpfulness evaluation? E.g., Anthropic's HH dataset or AlpacaEval?
>
> **A2**:  Thanks for your insightful question. First, we added the results on MT-bench [1] to evaluate the helpfulness of the fine-tuned LLMs. Moreover, we added more benchmarks, i.e. BBH [2] and HumanEval [3], to verify the capability of LLMs is not affected in most cases. The results are added in Table 1 and Table 2.
>
> [1] Judging LLM-as-a-judge with MT-Bench and Chatbot Arena.
>
> [2] Challenging BIG-Bench Tasks and Whether Chain-of-Thought Can Solve Them.
>
> [3] Evaluating Large Language Models Trained on Code.

---

> ### Author Response · Authors · 2023-11-19
>
> **Q3**: The title is misleading for two reasons.
>
> **A3**: Thanks for your helpful suggestions. We decided to change our title to “On the vulnerability of value alignment in open-access LLMs” and revise the claims in the paper to avoid possible misunderstandings. Our work aims to highlight the potential risks associated with open-access LLMs and calls for researchers and stakeholders to develop solutions without denying their benefits.
>
> Your suggestion of extending the scope to the fine-tuning API is quite reasonable and insightful. However, we currently decide to focus on investigating issues related to open-access models, for the following reasons:
> 1. While our HarmD and HelpD methods can be used for the fine-tuning API, other methods require more flexible model manipulation and thus can not be achieved by fine-tuning API. For example, HarmS and HarmQ involve removing optimization for the eos token, and RDPO requires training with the DPO method instead of SFT. These methods demonstrate the flexibility of attack strategies for open-access models.
> 2. The difficulty of defenses for the fine-tuning API and open-access LLMs is significantly different. Fine-tuning API on harmful dataset can be filtered out by service provider such as OpenAI, while fine-tuning open-access LLMs can be achieved without any restriction, which is more threatening as the attackers can obtain and manipulate the uncensored model in an offline, secretive, and unregulated manner. Please also refer to several strategies for responsible open-access LLM releases in the discussion section.
>
> ---
> **Ethics Concerns**:
>
> To address the ethical concerns, we added an ethical consideration section before references and discussed more potential solutions in the discussion section.

---

> ### Author Response · Authors · 2023-11-21
>
> Dear Reviewer dUB6,
>
> With the discussion period drawing to a close, we are eager to understand if our response has met your concerns. Any additional insights you may provide would be invaluable as we strive to refine our submission in these final days. We appreciate your time and guidance.
>
> After confirming with the chair and reviewing the updates to the [ICLR Author Guide](https://iclr.cc/Conferences/2024/AuthorGuide) on the 15th, we regret to learn that it is not possible to modify the paper's title and abstract in the system during this year's ICLR discussion stage. However, we have made changes in our revision to the best of our abilities to ensure that no misunderstandings occur.
>
> Best regards,
>
> Authors of Submission 2540

---

### Official Review · Reviewer_RY57 · 2023-11-01

**Soundness:** 3 good
**Presentation:** 3 good
**Contribution:** 3 good
**Rating:** 6
**Confidence:** 3

**Summary:**

This paper studies the vulnerability of value-aligned open-source LLMs to reverse alignment through fine-tuning. It proposes two fine-tuning strategies over various training data types with different difficulties of collection. Ultimately, they identify some successful strategies that can consistently reverse alignment while preserving the model's utility.

**Strengths:**

+ Open-source LLMs are getting better and better and their nefarious uses are becoming a concern. This paper shows that simple guardrails provided by value alignment are ineffective against fine-tuning.
+ A thorough investigation of different strategies of reverse fine-tuning.

**Weaknesses:**

- No exploration of automated, semantic jailbreak attacks [1,2] which might be a more common tool for adversaries [3]. Instead of fine-tuning, adversaries might prefer to use these jailbreaks, which are more straightforward and don't require collecting any training data. I recommend the authors to compare fine-tuning-based reversal to such jailbreak attacks as well.

- The differences between different LLMs are poorly explained. Baichuan2 model seems to be more vulnerable than Llama, and, although, there's some speculation in the paper ("the appropriate hyperparameter Beta for Lllama2-Chat is larger"), I would like to see a deeper exploration and explanation of these differences. For example, does more data or more aggressive fine-tuning against Llama equalize the results?

[1] https://arxiv.org/pdf/2202.03286.pdf
[2] https://arxiv.org/abs/2307.08715
[3] https://arxiv.org/abs/2308.03825

**Questions:**

See above.

---

> ### Author Response · Authors · 2023-11-19
>
> **W1**: No exploration of automated, semantic jailbreak attacks [1,2] which might be a more common tool for adversaries [3]. Instead of fine-tuning, adversaries might prefer to use these jailbreaks, which are more straightforward and don't require collecting any training data. I recommend the authors compare fine-tuning-based reversal to such jailbreak attacks as well.
>
> **A1**: Thanks for your helpful comments. Following your advice, we make the following two adjustments:
>
> 1. We add the results of manually-written Jailbreak prompts in Appendix A.2.2. The results show that the manually-written Jailbreak prompts are effective for Baichuan2-Chat but do not work for Llama2-Chat.
>
> 2. We also update the comparison in Section 5.3, where we compare reverse alignment with the state-of-the-art jailbreak attacks (hand-written jailbreaks for Baichuan, GCG for Llama2-Chat). From the results, we can see that most reverse alignment methods achieve **higher ASR than SOTA jailbreak attacks on the more robust Llama2-Chat**; while reverse alignment achieves comparable ASR with SOTA jailbreak attacks on the less robust Baichuan2-Chat,
>
> ---
>
> **W2**: The differences between different LLMs are poorly explained. Baichuan2 model seems to be more vulnerable than Llama, and, although, there's some speculation in the paper ("the appropriate hyperparameter Beta for Lllama2-Chat is larger"), I would like to see a deeper exploration and explanation of these differences. For example, does more data or more aggressive fine-tuning against Llama equalize the results?
>
> **A2**: In our experiments, we have several observations to support Baichuan2-Chat is more vulnerable than Llama2-Chat:
> Fine-tuning LLMs with benign datasets, i.e., LIMA and WizardLM, hurts the harmlessness of Baichuan2-Chat more than that of Llama2-Chat.
> RDPO is effective for Baichuan2-Chat, but does not work for Llama2-Chat.
> In the experiments of HarmD and HarmS, we find Llama2-Chat requires larger learning rates to achieve higher ASRs.
> Manually-written Jailbreak prompts can effectively attack Baichuan2-Chat, but fail to attack Llama2-Chat.
>
> These phenomena are summarized in Appendix A.2.3. We further added some analysis in Appendix A.2.3. We observe that Baichuan2-Chat (7B) has a much lower perplexity than the other three models. Moreover, Baichuan2-Chat handles alignment mostly in the word embedding layer, while Llama2-Chat handles alignment in the early Transformer layers. These could be reasons for their varying robustness.

---

> ### Author Response · Authors · 2023-11-21
>
> Dear Reviewer RY57,
>
> With the discussion period drawing to a close, we are eager to understand if our response has met your concerns. Any additional insights you may provide would be invaluable as we strive to refine our submission in these final days. We appreciate your time and guidance.
>
> Best regards,
>
> Authors of Submission 2540

---

### Official Review · Reviewer_X8UN · 2023-11-03

**Soundness:** 2 fair
**Presentation:** 3 good
**Contribution:** 2 fair
**Rating:** 5
**Confidence:** 4

**Summary:**

This paper focuses on the vulnerability of the safety alignment performed on open-source LLMs.  The authors propose the following two “reverse alignment” methods: (1)  fine-tuning these aligned LLMs with harmful datasets and the objective of maximizing the log-likelihood of targeted responses (2) fine-tuning the aligned LLMs to reverse direct preference optimization (DPO) to steer the preference to harmful responses. Both of them can be performed by parameter-efficient fine-tuning techniques. The experiments across different datasets demonstrated the effectiveness of proposed fine-tuning attacks.

**Strengths:**

1. The paper uncovers the vulnerability of safety alignment in terms of a new perspective (i.e., fine-tuning)
2. This paper focuses on two of them (i.e., SFT and DPO) and introduces corresponding attacks utilizing reverse fine-tuning with harmful datasets, which could serve as initial works on fine-tuning-based jailbreak to motivate the community work on more robust safety alignment or stealthier attacks that could bypass safety auditing.

**Weaknesses:**

1. The technical novelty of this paper is rather limited and the idea that fine-tuning can break existing alignment is really not something surprising.

2. The prepared dataset used to fine-tune is too broad to show the universality of the attack. For example, the authors fine-tune an aligned LLM with AdvBench while also evaluating the ASR on AdvBench. Even though they also evaluate its performance on other datasets, the fine-tuning dataset has contained all kinds of harmful scenarios, thus it is hard to demonstrate its universality on unseen harmful instructions.

3. The scope of this paper is the safety vulnerability of open-sourced LLMs. Recently, close-sourced LLMs such as GPT3.5 have also provided cloud-based fine-tuning services. It would be more impactful if the scope could be extended to close-sourced models as conducted in [1].

[1] Qi X, Zeng Y, Xie T, et al. Fine-tuning Aligned Language Models Compromises Safety, Even When Users Do Not Intend To![J]. arXiv preprint arXiv:2310.03693, 2023.

4. The performance is poor for the reverse DPO attack on Llama2-Chat?

**Questions:**

see above

---

> ### Author Response · Authors · 2023-11-19
>
> **W1**: The technical novelty of this paper is rather limited and the idea that fine-tuning can break existing alignment is not something surprising.
>
> **A1**: Besides investigating a range of methods to break alignment, the most significant aspect of our work is to expose the potential risks of open-access LLMs being easily reversely fine-tuned for outputting malicious content. Through the work, we advocate for more robust alignment methods for open-access LLMs. We have also discussed some potential solutions in the discussion section and plan to explore them in our future work.
>
> Moreover, our work also reveals that value alignment is not sufficient to eliminate the harmful knowledge that is embedded in the LLM’s weights. This knowledge can be reactivated with simple fine-tuning, even without introducing new harmful inputs. For example, in HarmS, the questions and prefixes are all generated by aligned LLMs. This means that we do not inject any additional harmful knowledge into the models.
>
> ---
>
> **W2**: The prepared dataset used to fine-tune is too broad to show the universality of the attack. Even though they also evaluate its performance on other datasets, the fine-tuning dataset has contained all kinds of harmful scenarios, thus it is hard to demonstrate its universality on unseen harmful instructions.
>
> **A2**: Thanks for your insightful suggestions. To address the concern, we have taken the following steps in the revision:
> We add a section (Section 5.4) to study the universality and transferability of reverse alignment. The results show that training on a set of harmful questions enables the LLM to answer unseen and dissimilar malicious questions.
> We mark the ASR results that overlap with the training set with an asterisk (*). We would like to clarify that not all fine-tuning datasets used in our experiments are broad. For example, WizardLM and LIMA do not contain harmful tasks. TDC only contains 50 harmful tasks.
>
> ---
>
> **W3**: The scope of this paper is the safety vulnerability of open-sourced LLMs. Recently, close-sourced LLMs such as GPT3.5 have also provided cloud-based fine-tuning services. It would be more impactful if the scope could be extended to close-sourced models as conducted in [1].
>
> **A3**: Thanks for your suggestions. The work in [1] is concurrent with ours and points out potential vulnerabilities in fine-tuning API. However, our work is still different from it in the following two key perspectives.
> 1. **Methodology**. While our HarmD and HelpD methods can be used for the fine-tuning API, other methods require more flexible model manipulations and thus can not be achieved by fine-tuning API. For example, HarmS and HarmQ involve removing optimization for the eos token, and RDPO requires training with the DPO method instead of SFT. These methods demonstrate the flexibility of attack strategies for open-access models.
> 2. **Difference of defenses**. The defenses for the fine-tuning API and open-access LLMs are significantly different. Fine-tuning API on harmful datasets can be filtered out by service providers such as OpenAI, while fine-tuning open-access LLMs can be achieved without any restriction, which is more threatening as attackers can obtain and manipulate the uncensored model in an offline, secretive, and unregulated manner. Please also refer to several strategies for responsible open-access LLM releases in the discussion section.
>
> [1] Fine-tuning aligned language models compromises safety, even when users do not intend to!
>
> ---
>
> **W4**: The performance is poor for the reverse DPO attack on Llama2-Chat?
>
> **A4**: We explain and analyze this phenomenon from two perspectives.
>
> First, several existing works have reported that Llama2-Chat is more robust and safer, which is also observed in our experiments. In Appendix A.2.3, we summarize the observations and provide a deeper analysis of the gradients and perplexity of different LLMs.
>
> Second, DPO (and RLHF) imposes a loss in controlling the consistency of the output distribution between the fine-tuned model and the original model, which ensures LLMs output fluent answers. Since the output probability of Llama2-Chat on harmful responses is very low, setting $\beta$ to a large value causes the model to be unable to output harmful responses. However, if the $\beta$ is set to a small value, Llama will not be able to output fluent sentences.

---

> ### Author Response · Authors · 2023-11-21
>
> Dear Reviewer X8UN,
>
> With the discussion period drawing to a close, we are eager to understand if our response has met your concerns. Any additional insights you may provide would be invaluable as we strive to refine our submission in these final days. We appreciate your time and guidance.
>
> Best regards,
>
> Authors of Submission 2540

---

> ### Author Response · Authors · 2023-11-23
>
> Dear reviwer X8UN,
>
> Thank you once again for your valuable efforts in reviewing our work. We have provided detailed responses to all of your comments and regret that we were unable to engage in a discussion with you during the rebuttal phase. As the deadline for rebuttal submissions is approaching, we sincerely anticipate your feedback. Here is a summary of our responses:
>
> 1. The novelty of this work: Our work aims to timely highlight the insufficiency of value alignment in preventing the **misuse of open-access LLMs**, which is a crucial and up-to-date topic nowadays. Despite alignment, these LLMs still **harbor malicious knowledge** that can be activated through various fine-tuning methods. Our work serves as a **whistleblower**, drawing the attention of researchers and the public to this issue and advocating for the development of more robust methods for open-accessing LLMs. We also discuss these more robust methods in the discussion section of our paper.
> 2. The universality of the attack: We add experiments to prove that training on a set of harmful questions enables the LLM to answer unseen and dissimilar malicious questions (**Section 5.4**).
> The difference to close-sourced models: First, the open-access LLMs offer **more flexible manipulation options** for attack strategies. Second, **defense mechanisms** for open-access LLMs and close-source APIs are different.
> 3. We explain the reason why RDPO does not work for Baichuan2-Chat by examining the differences in alignment between Llama2-Chat and Baichuan2-Chat (**Appendix A.2.3**), as well as the composition of the **DPO's loss**.
>
> Best regards,
>
> Authors of Submission 2540

---

### Author Response · Authors · 2023-11-19
**General Response**

We thank all the reviewers for their efforts in providing detailed comments. Our paper takes a preliminary step in revealing the misuse of open-access models. We are very grateful to all reviewers who agreed that this issue is a timely and important safety issue. Your opinions are very helpful for us to revise the paper. We also look forward to further discussions to help us better understand this issue in depth.


The main modifications in our revision are listed as follows. We also mark the modifications in blue in our revision.
1. We change the title and statement in our revision to avoid misunderstandings. We also send emails and messages to the chairs to help us change the title in the Open Review system.
2. We add an ethical consideration section.
3. We add several experiments, including:
   * Add transferability analysis to investigate the universality of reverse alignment (Section 5.4)
   * Add the results on more benchmarks to prove reverse alignment does not hurt the capability and helpfulness of reverse-aligned LLMs in most cases (Section 5.2).
   * Analyze the performance of reverse alignment on different types of harmful tasks (Appendix A.2.1)
   * Add experiments of manually-written jailbreak prompts (Appendix A.2.2).
   * Analyze the difference between Llama2-Chat and Baichuan2-Chat (Appendix A.2.3)
   * Add experiments of reverse aligning Llama2 (Appendix A.2.4).

---

> ### Author Response · Authors · 2023-11-23
> **Thanks for the Efforts of Reviewers and Chairs**
>
> Dear Reviewers and Chairs,
>
> We thank you again for your efforts and the insightful suggestions provided, which have significantly aided in refining our paper. We are also honored to receive feedback during the rebuttal phase indicating that our revisions have addressed some of your concerns. However, we regret the missed opportunity for interactive discussions with some reviewers during this phase to further resolve concerns. We understand and respect that your busy schedules may not allow for such discussions.
>
> As the deadline for the discussion phase approaches, we summarize our modifications made during the rebuttal phase again. We hope this summary will be helpful in your post-discussion phase deliberations.
>
> 1. We change the title and statement in our revision to avoid misunderstandings. We also send emails and messages to the chairs to help us change the title in the Open Review system.
> 2. We add an ethical consideration section.
> 3. We add several experiments, including:
>     * Transferability analysis to investigate the universality of reverse alignment (Section 5.4)
>     * Results on more benchmarks to prove that the reverse alignment does not hurt the capability and helpfulness of reverse-aligned LLMs in most cases (Section 5.2).
>     * Analyzing the performance of reverse alignment on different types of harmful tasks (Appendix A.2.1)
>     * Experiments of manually-written jailbreak prompts (Appendix A.2.2).
>     * Analyzing the difference between Llama2-Chat and Baichuan2-Chat (Appendix A.2.3)
>     * Experiments of reverse aligning Llama2 (Appendix A.2.4).
>
>
> Thanks,
>
> Authors of Submission 2540

---

### Meta-Review · Area_Chair_hFhh · 2023-12-06

**Metareview:**

This paper studies the danger of open-sourced LLMs in terms of value alignment. The selected topic is important and tries to give an initial answer on the recent debate about whether LLMs should be open-sourced. However, reviewers have concerns on the technical novelty of the paper and find that the key message of "fine-tuning can break alignment" is a known property of LLMs as forgetting. AC encourages the authors to take these reviews into account to further polish the study for a future submission.

**Justification For Why Not Higher Score:**

Many reviewers have concerns on the technical novelty of the paper after the rebuttal phase.

**Justification For Why Not Lower Score:**

N/A

---

### Decision · Program_Chairs · 2024-01-16

Reject